# Saquinavir-Piperine Eutectic Mixture: Preparation, Characterization, and Dissolution Profile

**DOI:** 10.3390/pharmaceutics15102446

**Published:** 2023-10-11

**Authors:** Cinira Fandaruff, María Isabel Quirós-Fallas, José Roberto Vega-Baudrit, Mirtha Navarro-Hoyos, Diego German Lamas, Andrea Mariela Araya-Sibaja

**Affiliations:** 1Instituto de Tecnologías Emergentes y Ciencias Aplicadas (ITECA), UNSAM-CONICET, Escuela de Ciencia y Tecnología, Laboratorio de Cristalografía Aplicada, Av. 25 de Mayo 1169, San Martín 1650, Provincia de Buenos Aires, Argentina; dlamas@unsam.edu.ar; 2Laboratorio Nacional de Nanotecnología, LANOTEC-CeNAT-CONARE, San José 1174-1200, Costa Rica; maria.quirosfallas@ucr.ac.cr (M.I.Q.-F.); jvegab@gmail.com (J.R.V.-B.); 3Laboratorio Biodess, Escuela de Química, Universidad de Costa Rica, San Pedro de Montes de Oca, San José 2060, Costa Rica; mnavarro@codeti.org

**Keywords:** saquinavir, piperine, eutectic mixtures, solid-state characterization, powder diffraction, small-angle X-ray scattering, dissolution enhancement

## Abstract

The dissolution rate of the anti-HIV drug saquinavir base (SQV), a poorly water-soluble and extremely low absolute bioavailability drug, was improved through a eutectic mixture formation approach. A screening based on a liquid-assisted grinding technique was performed using a 1:1 molar ratio of the drug and the coformers sodium saccharinate, theobromine, nicotinic acid, nicotinamide, vanillin, vanillic acid, and piperine (PIP), followed by differential scanning calorimetry (DSC). Given that SQV-PIP was the only resulting eutectic system from the screening, both the binary phase and the Tammann diagrams were adapted to this system using DSC data of mixtures prepared from 0.1 to 1.0 molar ratios in order to determine the exact eutectic composition. The SQV-PIP system formed a eutectic at a composition of 0.6 and 0.40, respectively. Then, a solid-state characterization through DSC, powder X-ray diffraction (PXRD), including small-angle X-ray scattering (SAXS) measurements to explore the small-angle region in detail, Fourier transform infrared spectroscopy (FT-IR), scanning electron microscopy (SEM), and a powder dissolution test were performed. The conventional PXRD analyses suggested that the eutectic mixture did not exhibit structural changes; however, the small-angle region explored through the SAXS instrument revealed a change in the crystal structure of one of their components. FT-IR spectra showed no molecular interaction in the solid state. Finally, the dissolution profile of SQV in the eutectic mixture was different from the dissolution of pure SQV. After 45 min, approximately 55% of the drug in the eutectic mixture was dissolved, while, for pure SQV, 42% dissolved within this time. Hence, this study concludes that the dissolution rate of SQV can be effectively improved through the approach of using PIP as a coformer.

## 1. Introduction

Improving the efficacy of Active Pharmaceutical Ingredients (APIs) belonging to Class II and IV in the Biopharmaceutical Classification System (BCS) is a challenge for pharmaceutical industries since approximately 70–90% of pharmaceutical compounds present poor aqueous solubility and, thus, inadequate bioavailability [1]. Over 50 years ago, crystal engineering was applied to a wide range of crystalline powders in order to modulate the performance of Class II and IV raw materials [2,3,4]. Since then, the process of reducing particle size to increase the surface area [5,6,7], as well as obtaining salts [4,8,9], nanocrystals [10,11], cocrystals [12,13], and eutectic mixtures [14,15,16] were some of the approaches that have been explored.

Saquinavir base (SQV) (Figure 1) is an anti-HIV drug which belongs to Class II in the BCS. It presents low aqueous solubility and extremely low absolute bioavailability, less than 4% [17]. Therefore, strategies involving processing or formulation modifications are desired to enhance its oral bioavailability, addressing a relevant shortcoming for new strategies to improve SQV bioavailability. In fact, few studies on formulation modifications have been published related to SQV. Surjyanarayan and colleagues developed an oral microemulsion based on a drug delivery system for enhancing the bioavailability of SQV [18]. Another work involving cyclodextrin for improving oral absorption was carried out by Pathak and colleagues [17]. In both cases, the success of the techniques is dependent upon the physicochemical characteristics of the drug. Accordingly, a strategy involving process modification for SQV through the obtainment of a eutectic mixture was attempted in this paper. Herein, a natural product, Piperine (PIP) (Figure 1), has been applied as a coformer in order to obtain multicomponent organic materials. In a previous study, the solubility and dissolution rate of Class II drugs—curcumin, lovastatin and irbesartan—were enhanced by PIP through eutectic mixtures [19]. PIP has been used as a “bioenhancer” and inhibits CYP3A4 and P-glycoprotein, as well as the glucuronidation pathway. This pathway is the most important reaction of reducing xenobiotics/ drugs and, thus, also the pre-systemic metabolism of some drugs belonging to Class II and IV in the BCS [20]. Furthermore, PIP presents low toxicity in both humans and animals and is generally regarded as a safe (GRAS) substance [19].

An eutectic mixture is defined as a system where the melting point of the mixture is below the melting point of each isolated component [21,22]. Over the last six decades, plenty of reports were published describing the obtention of eutectic mixtures, including low aqueous solubility APIs and hydrophilic agents, with the purpose of improving dissolution drugs [23]. Interestingly, these mixtures are still under preliminary development and represent a novelty in terms of products on the pharmaceutical market, probably due to the lack of literature regarding their molecular structure, bonding interactions, and crystalline organization [24,25]. The well-known eutectic composition of the local anesthetic drugs lidocaine and prilocaine (Emla^®^) is currently used to enhance the transdermal permeation of lidocaine [24,26]. Although the mechanism by which the dissolution/solubility of eutectic systems increases is not completely understood, some explanations have been reported. The use of highly water-soluble carriers and the better wettability in the microenvironment created between the drug particles and the hydrophilic component leading to local solubilization effects are some highlighted factors. In this case, crystal packing, molecular mobility, and intermolecular interaction can be modulated [22,25,27]. Other factors include particle size reduction, absence of aggregation and agglomeration between hydrophobic drug particles, and appropriate wettability and dispersibility of the mixture [25].

Eutectic mixtures are easily produced [24] and may offer a potentially fruitful alternative for combination therapy [16]. Furthermore, the eutectic mixtures are cost-effective, easily scaled-up, and do not require clinical trials [16,28]. Hence, the aim of this study was to obtain a eutectic mixture from SQV using PIP as a coformer to improve the dissolution of this Class II drug. Differential scanning calorimetry (DSC) was applied to conduct thermal analysis. The optimal compositions for SQV-PIP were identified according to the binary phase and Tammann diagrams. The eutectic system was characterized through scanning electron microscopy (SEM), fourier transform infrared spectroscopy (FT-IR); powder X-ray diffraction (PXRD), and small-angle X-ray scattering (SAXS) to explore the small-angle region of the diffractograms with high resolution. Even though this methodology is not often employed for the solid-state characterization of pharmaceutical compounds, in the present case, it proved to be very powerful in revealing Bragg peaks that could otherwise be missed due to a high degree of peak overlapping. Finally, a powder dissolution test was performed to determine the dissolution enhancement of SQV.

## 2. Materials and Methods

### 2.1. Materials

SQV was provided by Cristália Produtos Químicos Farmacêuticos Ltda (São Paulo, Brazil) and used without further purification. Sodium saccharinate, theobromine, nicotinic acid, nicotinamide, vanillin, vanillic acid, and piperine were purchased from Sigma Aldrich from Laramie, WY, USA. All other materials and solvents used were analytical-grade reagents.

### 2.2. SQV Eutectic Mixture Screening

SQV and sodium saccharinate, theobromine, nicotinic acid, nicotinamide, vanillin, vanillic acid, and piperine (PIP) were individually and accurately weighed to obtain 50 mg of a binary mixture in a 1:1 molar ratio. The powders were ground together, assisted with 50 µL of ethanol in a glass mortar and a pestle for around 5 min, or until the solid was entirely dried. The obtained solids were dried at 60 °C for 2 h and stored in a desiccator until further analysis. Each mixture was submitted to the DSC analyses to prove the eutectic formation. Then, approximately 2 mg of the mixture was placed into an aluminum crucible and analyzed from 40 °C to 200 °C or 300 °C, depending on the coformers’ melting point at a heating rate of 10 °C/min, under a dynamic nitrogen atmosphere of 50 mL/min using the DSC equipment described in Section 2.4.1.

### 2.3. SQV-PIP Eutectic System

#### 2.3.1. Determination of Mixtures Composition at the Eutectic Point

Through the construction of Tammann’s and binary phase diagrams, the composition at the eutectic point of the SQV-PIP mixture was obtained. The diagrams were constructed according to Chadha and colleagues [29] and as recommended by Rycerz [30]. Different molar ratios of SQV and PIP of 0.9:0.1, 0.8:0.2, 0.7:0.3, 0.6:0.4, 0.5:0.5, 0.4:0.6, 0.3:0.7, 0.2:0.8, and 0.1:0.9 were prepared as described in Section 2.2. Subsequently, 2 mg of the produced solid was weighed into an aluminum crucible and analyzed from 40 to 200 °C, as described in Section 2.2. In the phase diagram construction, the onset temperature of the first endothermic event was used as the solidus point, and the peak of the second endothermic event was considered the liquidus point. In the Tammann diagrams, the onset temperature, and the enthalpy of fusion of the first endothermic event were used. TA Instruments-Waters LLC Universal Analysis 2000 software (version 4.5A, New Castle, DE, USA, 2016) was used for performing data analyses. The samples were analyzed in triplicate.

#### 2.3.2. Preparation of Bulk Mixtures at the Eutectic Composition

The SQV-PIP eutectic system was prepared in a large amount in the respective eutectic composition to carry out its solid-state characterization. The method mentioned previously in Section 2.2 was scaled-up. Briefly, 500 mg SQV and 320 mg PIP were accurately weighted to obtain the SVQ-PIP system (0.6:0.40). The liquid-assisted grinding was used to prepare the eutectic mixture that was homogenized in a glass mortar and a pestle using 400 µL of ethanol until dry. The obtained solids were dried at 60 °C for 2 h and stored in a desiccator until further analysis.

### 2.4. Solid-State Characterization of the Eutectic Mixtures

#### 2.4.1. Differential Scanning Calorimetry (DSC) Analysis

DSC analysis of eutectic mixtures was carried out using a DSC-Q200 (TA Instrument, New Castle, DE, USA) equipped with a TA Refrigerated Cooling System 90. The measurements were performed under the conditions described in Section 2.2.

#### 2.4.2. Powder X-ray Diffraction (PXRD)

The PXRD analyses were carried out in a PANalytical Empyrean diffractometer with a copper tube (λ = 1.548 Å), 45 KV, 40 mA and scanning from 4° to 45° (2θ) with a step size of 0.02 and a step time of 8 s. The diffractograms were obtained at ambient conditions.

The small-angle 2θ region of the diffractograms was explored in more detail by using a XENOCS SAXS instrument, Xeuss 2.0 model (Laboratorio de Cristalografía Aplicada, Instituto de Tecnologías Emergentes y Ciencias Aplicadas, ITECA, CONICET-UNSAM, Argentina). This instrument has a microfocus Cu-Ka GENIX 3D X-ray tube of ultra-low divergence and two Dectris Pilatus 2D detectors that can be used simultaneously: a 100 K detector for wide angles (WAXS technique) and a 200 K one for small angles (SAXS) or ultra-small angles (USAXS). The sample-detector distance can be varied between 10 cm and 6 m, changing the 2q range that is accessible. In these experiments, we took advantage of two important additional features of this instrument to achieve accurate data for 2q angles up to 10°: (1) The SAXS detector is motorized, so it can be converted in a larger detector (a measurement mode called “virtual detector”), thus reaching higher angles without losing resolution and (2) the X-ray beam can be highly collimated, with a minimum spot size of approximately 250 mm × 250 mm in the ultra-high resolution (UHR) mode. The UHR SAXS data shown below were measured under these conditions with a sample-detector distance of about 1200 mm, composing 18 images taken for 30 min each. It is also worth mentioning that the samples were measured in transmission geometry and carefully prepared to avoid any possibility of preferred orientation.

#### 2.4.3. Fourier Transform Infrared (FT-IR)

FT-IR analyses were performed on a Thermo Scientific Nicolet 6700 FT-IR (Waltham, MA, USA) spectrometer fitted with a diamond attenuated total reflectance (ATR) accessory. The FT-IR spectra were collected in the range of 4000 to 600 cm^−1^ using 32 scans at 4 cm^−1^. The samples were placed into the ATR cell and no further preparation was necessary.

#### 2.4.4. Scanning Electron Microscopy (SEM)

The morphology and crystal size of SQV were determined using a Philips (Amsterdam, The Netherlands), Model XL 30 microscope at a voltage of 15 kV. The samples were mounted on metal stubs using double-sided adhesive tape, vacuum-coated with gold (350 Å) in a Polaron E-5000. For PIP and SQV-PIP the measurements were collected in a JEOL JSM-6390 LV and operated with an acceleration voltage of 20 kV. The samples were mounted as previously described for the SQV sample, but in an argon atmosphere.

### 2.5. Powder Dissolution Test

The dissolution study was carried out for powder samples of pure SQV and the SQV-PIP eutectic mixture in their exact eutectic composition using a method based on the United States Pharmacopeia (USP) for SQV capsules on a SOTAX S7 dissolution test system. The dissolution media was a citrate buffer composed of anhydrous dibasic sodium phosphate and citric acid monohydrate dissolved in and diluted with water. A total of 200 mL of medium previously heated at 37.0 ± 0.5 °C was used and the experiment was carried out under 75 rpm stirring. Then 5 milliliters of the samples were withdrawn at 5, 15, 30, and 45 min. The samples were filtered, properly diluted, and analyzed. The test was performed in triplicate.

#### Determination of Drug Content

To determine the drug content in the dissolution test, a first-order derivative spectrophotometric method was developed and validated. All spectral measurements were carried out in 1 cm quartz cells using a Shimadzu UV 1800 double-beam recording spectrophotometer. Data were treated using the UVProbe software (2.42 Version). This system permits the derivative processing of spectra. The quantification was performed by the zero-crossing technique using the absorbance at 245 nm of the first-order derivative spectra.

## 3. Results and Discussion

### 3.1. Eutectic Mixture Screening

Despite the great advances in in-silico tools to determine intermolecular interactions capable of forming a eutectic mixture, there is no other approach to identify eutectic formation than trial and error using DSC measurements of a binary mixture. Therefore, DSC is an important technique applied to identify the eutectic system. A marked decrease in the melting temperature of the eutectic mixture when compared with its parent compounds occurred when the mixture was submitted to thermal analysis [31]. Regarding the lower temperature of the eutectic mixture compared to the starting materials in DSC, the strong hydrogen bonding interactions are responsible for the important lowering of the melting point in the eutectic system [32]. DSC curves of individual component SQVs and PIPs as well as the 1:1 binary mixture are shown in Figure 2. As observed, there was a unique endothermic event which took place at a lower temperature with respect to the starting materials, which makes evident the formation of the SQV-PIP eutectic mixture. In this regard, SQV and PIP have sufficient intermolecular physical interacting groups for eutectic obtainment [33,34] that could contribute to obtaining a eutectic mixture between these two components. No eutectic mixture formation was observed between SQV and the other coformers sodium saccharinate, theobromine, nicotinic acid, nicotinamide, vanillin, and vanillic acid (DSC curves not shown).

Consequently, the composition at the eutectic point, the solid-state characterization and the dissolution study were carried out on the SQV-PIP system.

### 3.2. Binary Phase and Tammann Diagrams

An appropriate determination of eutectic composition demands the construction of the Binary phase and Tammann diagrams [30]. Both were acquired for the eutectic mixture through DSC data from a series of melting endotherms of the binary mixtures obtained in variable molar ratios of SQV and PIP (0:1, 0.1:0.9, 0.2:0.8, 0.3:0.7, 0.4:0.6, 0.5:0.5, 0.6:0.4, 0.7:0.3, 0.8:0.2, 0.9:0.1, and 1:0 mol/mol), as shown in Figure 3a. A single melting endotherm was observed at a 0.4 molar ratio of SQV. At 0.8, 0.7, and 0.6 and 0.5 molar ratios of SQV, two melting endotherms were observed. For the phase diagram (see Figure 3b), the melting temperature of the eutectic mixture (solidus point) was plotted as a function of the mole fraction of PIP. Regarding the Tammann diagram, it showed the systematic dependence of molar enthalpy associated with the eutectic effect on the molar fraction [30]. In Figure 3c, the Tammann diagram shows the intercept point, which indicates the precise mole fraction of SQV: PIP in the eutectic composition, that is, 0.4:0.6 molar ratio.

### 3.3. Solid-State Characterization of SQV-PIP Eutectic System

A eutectic system consists of at least two solid components mixed at a specific molar ratio that exhibits an important decrease in the melting temperature in comparison with the temperature of each individual component [19,35,36]. The hydrogen bonds established between the components are responsible for the decrease in the melting point of the mixture [36]. The SQV molecule offers five hydrogen bond donors and seven hydrogen bond acceptors whereas PIP has three hydrogen bond acceptors. There is no hydrogen bond donor in the PIP molecule. SQV and PIP have sufficient intermolecular physical interacting groups for eutectic obtainment [33,34]. For these systems, the crystal structure remains unchanged from the individual components, in addition to the disorder introduced mainly due to the grinding during the system obtainment [16,19]. In order to obtain eutectic systems, neither new chemical entities nor new crystalline phases were expected. Therefore, a strategy for proving the eutectic system obtainment was to characterize them through appropriate techniques. Hence, to analyze the crystalline state and molecular interaction in the SQV-PIP eutectic system, a solid-state characterization was performed as presented in the next sections.

#### 3.3.1. PXRD Analyses

PXRD provides important information for the study of eutectic mixtures, since it allows us to detect crystal structure modifications [19,27]. The PXRD patterns determined for SQV, PIP, and the eutectic system presented in Figure 4 demonstrate that the individual components, as well as the system obtained, are crystalline. The PXRD pattern of the eutectic mixture maintains the characteristic peaks of SQV and PIP without a change in peak position, indicating no formation of a new crystalline phase. Decreases in reflections’ intensity observed in all diffractograms of eutectic-mixture occur due to amorphization of the system caused by the grinding process. For the small-angle region, there is a change in the peak shape and the position of SQV when compared with those of the eutectic mixture. In this last sample, the peaks at 4.6° and 5.3° are not well defined, while a new peak at 4.9° is detected. In addition, there is a marked change in the relative intensity of these small-angle peaks. The most relevant peaks of SQV at 9.3°, 16.1°, and 18.2° are present in the eutectic system, as well as the characteristics peaks of PIP at 12.9°, 14.7°, 19.7°, 21.3°, 22.6°, 25.8°, and 28.2°.

The small-angle region of the above diffractograms was explored in more detail by means of a SAXS instrument. In contrast to the usual case of the conventional diffractometers (with Bragg-Brentano geometry), these experiments were conducted in transmission geometry for very small samples, which allows a simpler sample preparation, avoiding problems of preferred orientation. Figure 5 shows the 2D diffraction patterns obtained using an ultra-high resolution (UHR) configuration for 2q ≤ 10°. Continuous Debye rings of uniformed intensities are clearly observed in both SQV and eutectic mixture samples, showing that the effects of preferred orientation can be neglected. No rings were detected in the PIP sample, which was expected, considering that it does not exhibit small-angle Bragg peaks in Figure 4. Since the 2D patterns shown in Figure 5 are isotropic, they can be azimuthally integrated and thus converted into conventional 1D diffractograms. The resulting data are shown in Figure 6 for 2q angles in the range of 2–7°.

As can be noticed in Figure 6, the UHR experiment resolved the Bragg peaks close to 2q = 5°. Now it can be clearly observed that the SQV sample exhibits two peaks of similar relative intensities, while the eutectic mixture presents three peaks. Two of these three peaks (at 2q = 4.75° and 5.30°) can be attributed to the SQV peaks; however, it is clear that the eutectic mixture exhibits an additional high-intensity peak at 2q = 4.97°. This reflection did not correspond to any of the already reported PIP [37] or SQV polymorphic forms [38]. These results indicate that, in this particular case, the eutectic mixture is not only a simple combination of the original phases, since there is a clear indication of a change in the crystal structure. Therefore, it could be due to the phase transition of one of the components to a new phase. However, to determine which of the components is undergoing a change, the SQV-PIP eutectic system needs to be closely studied and it will be the focus of a future contribution.

#### 3.3.2. FT-IR Analyses

FT-IR is a useful solid-state characterization technique to detect molecular interactions in the solid state commonly observed between drug and coformer, such as intermolecular hydrogen bonds [19,27]. The FT-IR spectra of SQV, PIP, and the eutectic mixture SQV-PIP are presented in Figure 7. The FT-IR of SQV showed characteristic bands at 1654 cm^−1^ related to C=O stretching and 1625 cm^−1^ for NH bending [39]. For PIP, the FT-IR showed bands at 1632 cm^−1^ assigned to (-C=O) stretching; at 1582 cm^−1^ related to the asymmetric bend of the carbonyl group (O=C-N); at 1492 cm^−1^ associated with aromatic (-C=C-) stretching; at 1441 cm^−1^ related to –CH_2_ bending; and at 1253 cm^−1^ associated with –C-O stretching [19]. The SQV and PIP characteristic peaks are presented in the SQV-PIP eutectic mixture. In the eutectic mixture, the SQV-PIP spectrum does not exhibit considerable shifting for the main functional group. The FT-IR and PXRD-associated results confirmed that no molecular interactions in the solid state took place and the eutectic system was formed.

#### 3.3.3. Scanning Electron Microscopy

The micrographs of the API, the coformer and the eutectic mixture are shown in Figure 8 as SQV (a and b), PIP (c and d), and the eutectic system (e and f). As observed in Figure 8, the eutectic system has a different morphology in comparison with the individual components. As previously described by Heinz and colleagues (2008), SQV shows agglomerates of fused rod-like particles with at least two populations of different particle sizes, the smallest measuring approximately 1–2 µm [40]. PIP shows tabular closely spaced morphology, as mentioned by Wilhelm-Romero and colleagues [19]. SQV-PIP present significantly different morphology compared to SQV and PIP individually.

In terms of size, SQV and SQV-PIP present smaller crystals in comparison to PIP crystals, which presents considerably larger crystal size than SQV and SQV-PIP ones. Therefore, different magnifications were used for SEM images. It is noteworthy that, due to the grinding process, the eutectic system presents a decrease in particle size.

#### 3.3.4. Powder Dissolution Test

SQV, as a Class II drug, presents a good correlation between in vivo results and dissolution tests and, therefore, the dissolution rate is the aspect to be controlled since it determines the drug absorption [41,42]. The dissolution medium chosen for SQV and SQV-PIP was citrate buffer, composed of anhydrous dibasic sodium phosphate and citric acid monohydrate, as indicated by USP for SQV capsules. The dissolution profile of the pure powdered compound and the eutectic mixture is shown in Figure 9. For the first 5 min, the percentage dissolved was quite similar for SQV compared to the eutectic mixture. The dissolution profile of SQV in the eutectic mixture was different from that of the pure SQV. After 45 min, approximately 55% of the drug in the eutectic mixture dissolved while, for pure SQV, 42% dissolved within the same time. It is worth mentioning that, despite the significant improvement of SQV dissolution in the SQV-PIP eutectic mixture, it does not show a burst dissolution behavior.

Even though the role of the coformer in the eutectic mixture for achieving higher dissolution compared to the pure drug is not well understood, there are some hypotheses based on evidence in scientific literature. The PIP coformer is a hydrophobic molecule and, according to the literature, some aspects, such as particle size reduction, dispersibility [43], absence of aggregation and agglomeration between the hydrophobic drug particles, and good wettability of the mixture, might contribute to the increase in the dissolution rate [43,44]. Additionally, the improvement of eutectic dissolution has been attributed to the weaker crystalline nature of the eutectic system. The weakness of the crystalline structure results from the molecular organization of the components in which there was a heterogeneous structural organization, which results in higher thermodynamic parameters [45]. In the eutectic system, for example, the crystal lattice is not as well packed as in the cocrystal and, therefore, a negligible barrier corresponding to crystal lattice interactions exists [46].

In addition, according to Lee and colleagues, PIP has an important effect on the pharmacokinetics and bioavailability of co-administered drugs and improved the bioavailability of nevirapine, another anti-HIV, in addition to drugs such as midazolam, propranolol, carbamazepine, theophylline, phenytoin, and curcumin [47]. Moreover, PIP seems to modulate the intestinal permeability and the expressions of P-glycoprotein (P-gp). Due to its apolar nature and the configuration it assumes with other molecules in the membrane, PIP affects the fluidity of the cellular membrane [48]. An increase in the membrane fluidity of epithelial cells may contribute to the enhanced absorption of drugs, such as salicylates and medroxyprogestrone, in epithelial and cancer cells, as demonstrated by Kajii and colleagues [48,49].

Considering the mentioned above, the fact that PIP presents low toxicity in both humans and animals and is classified as a GRAS molecule [19], together with the point that eutectic mixtures will not require clinical trials for its use [16,28], these results showed promising potential of pharmaceutical formulations. In addition, it would be interesting to continue studying this eutectic system by, for example, conducting toxicological tests.

## 4. Conclusions

SQV was able to form a eutectic system with PIP through liquid liquid-assisted grinding approach. Solid-state characterization of the bulk material suggested the absence of molecular interactions nor structural changes; however, the small-angle region explored in more detail through the SAXS instrument revealed changes in the crystal structure of one of their components which could not be detected by conventional PXRD measurement. The eutectic composition was calculated through both binary phase’s and Tammann’s diagrams. The obtained material led to an improved dissolution rate of SQV. Considering this enhancement and the fact that PIP is a natural, non-toxic, bioenhancer compound, and the accessibility of the method for the pharmaceutical industry, our results indicate that these SQV eutectic systems have promising applications in the preparation of new pharmaceutical formulations.

## Figures and Tables

**Figure 1 pharmaceutics-15-02446-f001:**
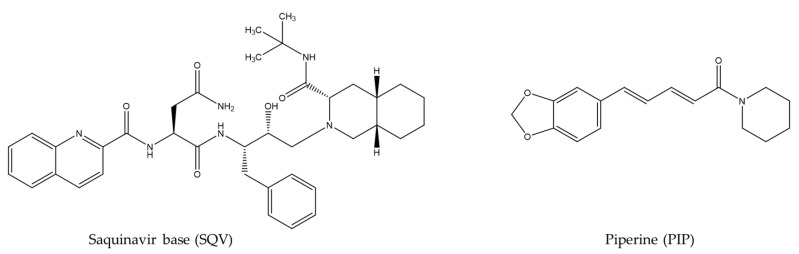
Chemical structures of saquinavir base (SQV) and piperine (PIP).

**Figure 2 pharmaceutics-15-02446-f002:**
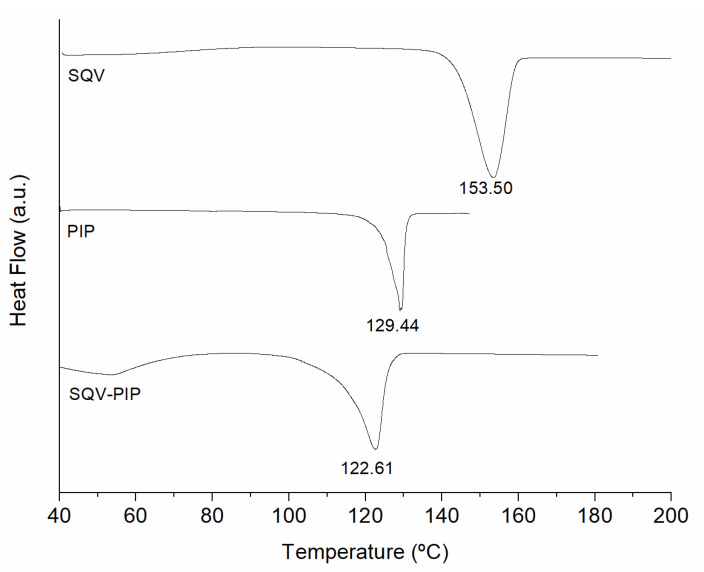
DSC analysis of the eutectic mixture screening. The DSC curves of SQV, PIP and the eutectic mixture SQV-PIP are presented.

**Figure 3 pharmaceutics-15-02446-f003:**
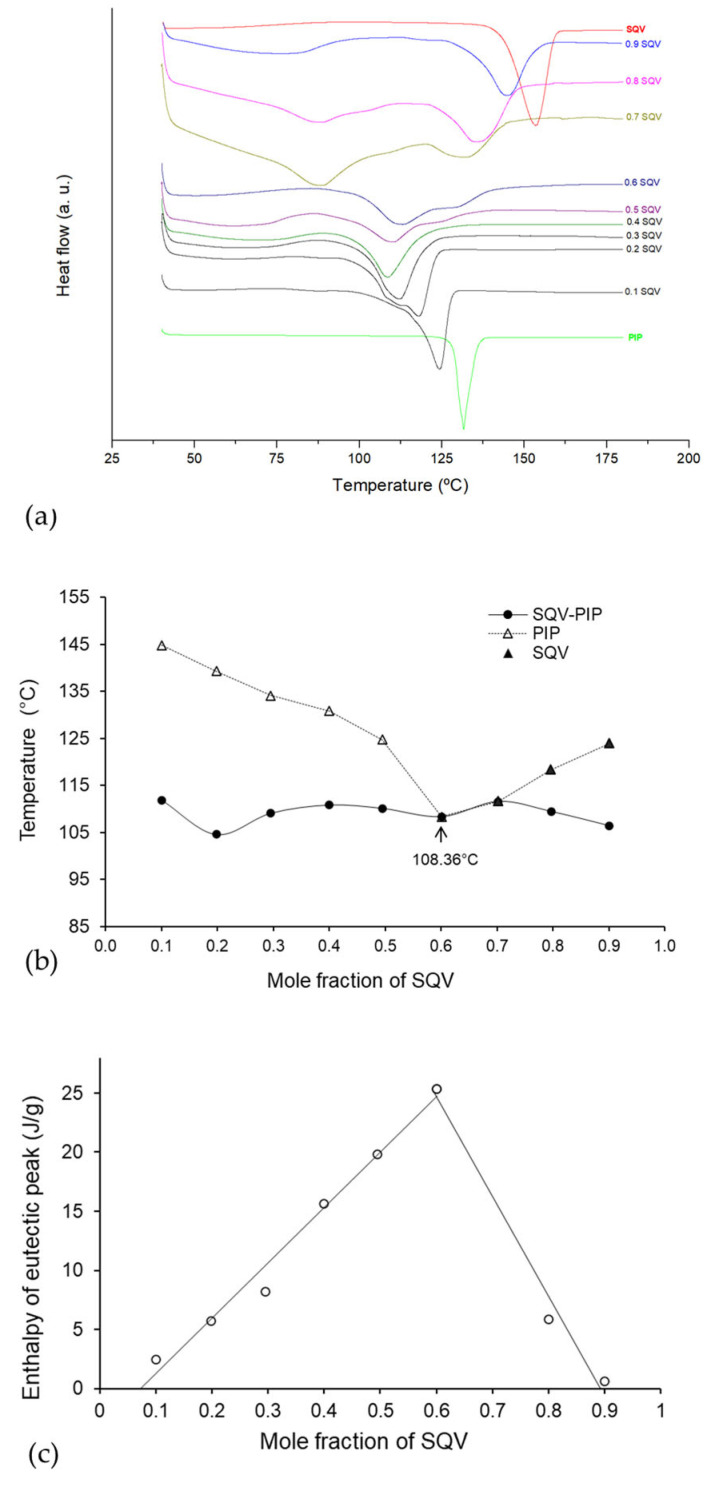
(**a**) DSC curves used to construct (**b**) the eutectic phase diagram where the filled triangle represents the variable liquidus line; the filled circles represent the solidus line. (**c**) The Tammann diagram of SQV-PIP eutectic mixture. Measurements to construct the phase and Tammann diagram were performed in triplicate (*n* = 3).

**Figure 4 pharmaceutics-15-02446-f004:**
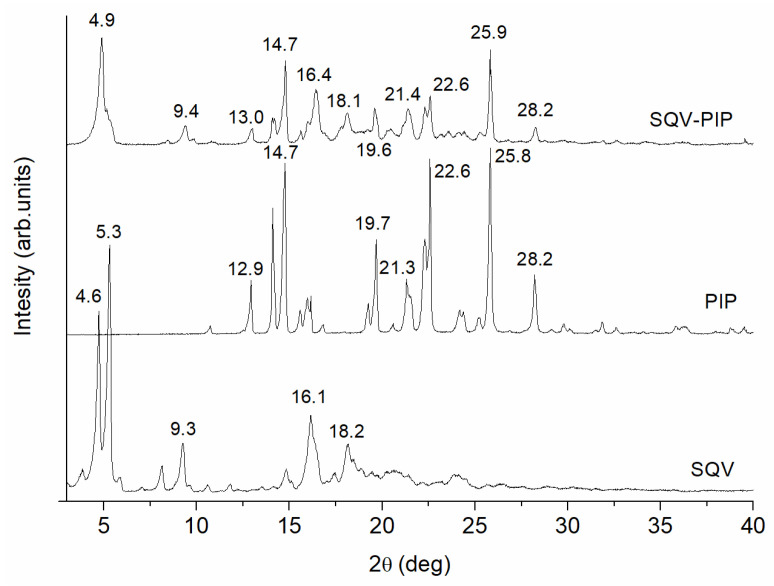
PXRD of the pure components as well as the eutectic system SQV-PIP at 0.4:0.6 molar ratio. Experiments were performed in replicate for each sample (*n* = 2).

**Figure 5 pharmaceutics-15-02446-f005:**
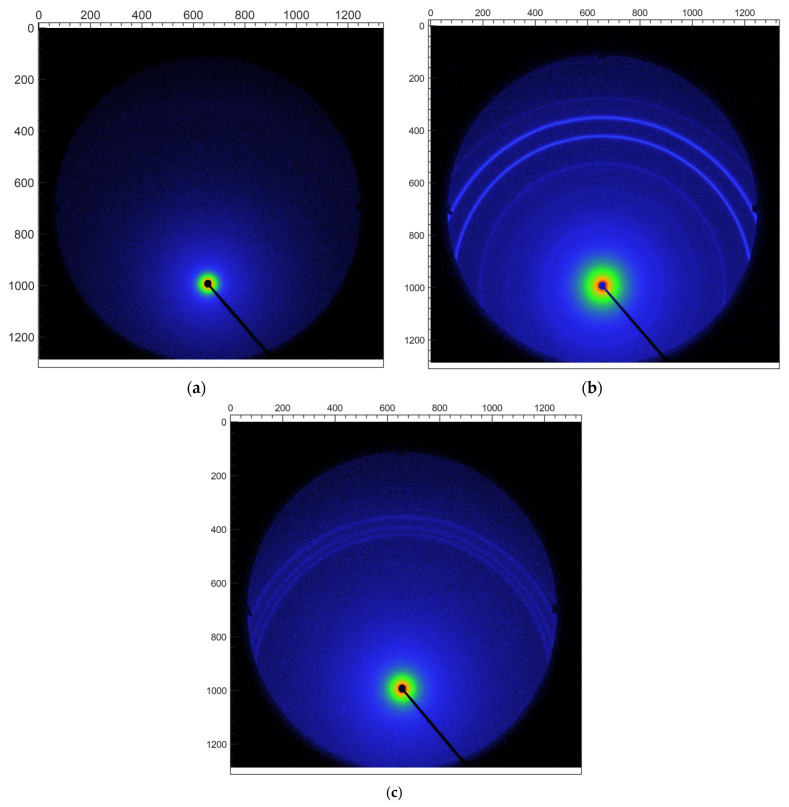
2D small-angle PXRD patterns of PIP (**a**), SQV (**b**), and SQV-PIP eutectic mixture (**c**) obtained using a SAXS instrument in an ultra-high resolution configuration. The region observed in these patterns corresponds to 2q angles up to 10°.

**Figure 6 pharmaceutics-15-02446-f006:**
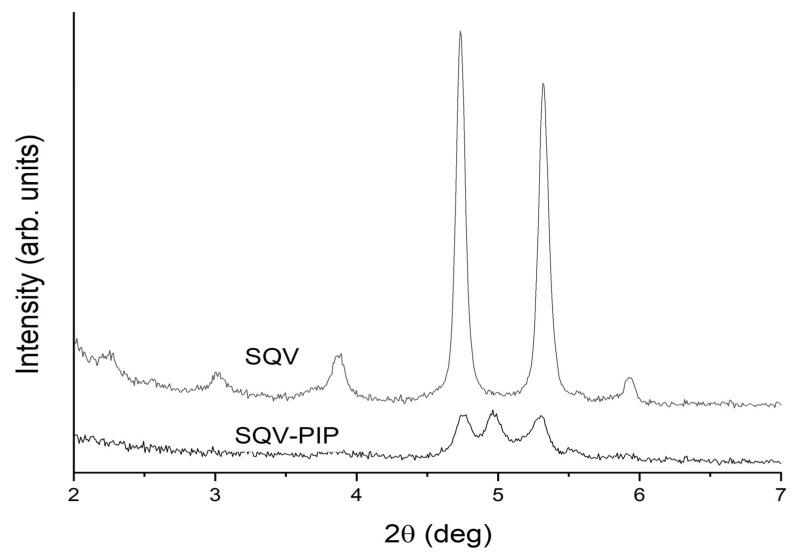
Small-angle PXRD data of the SQV and SQV-PIP (eutectic mixture) samples, obtained by azimuthal integration of the corresponding 2D patterns shown in Figure 5.

**Figure 7 pharmaceutics-15-02446-f007:**
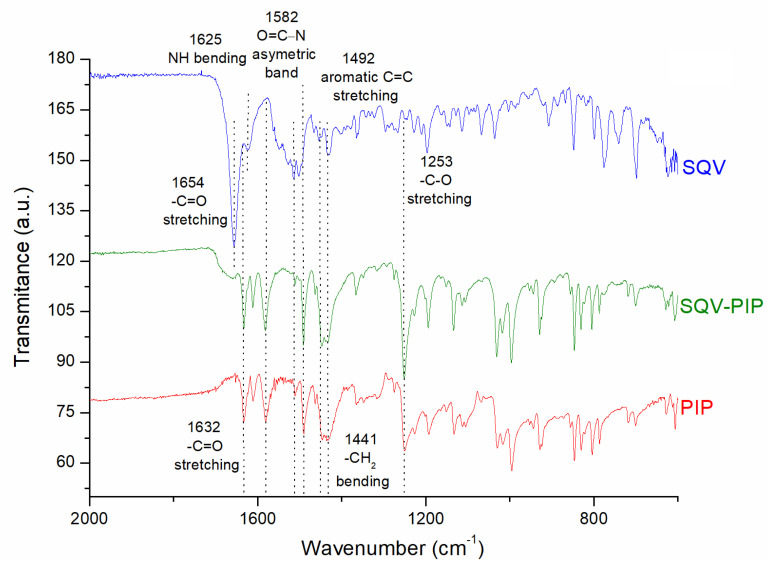
FT-IR spectra in the range of 600 to 2000 cm^−1^ of SQV, SQV-PIP eutectic system, and PIP. The dotted lines are used for indicating the bands.

**Figure 8 pharmaceutics-15-02446-f008:**
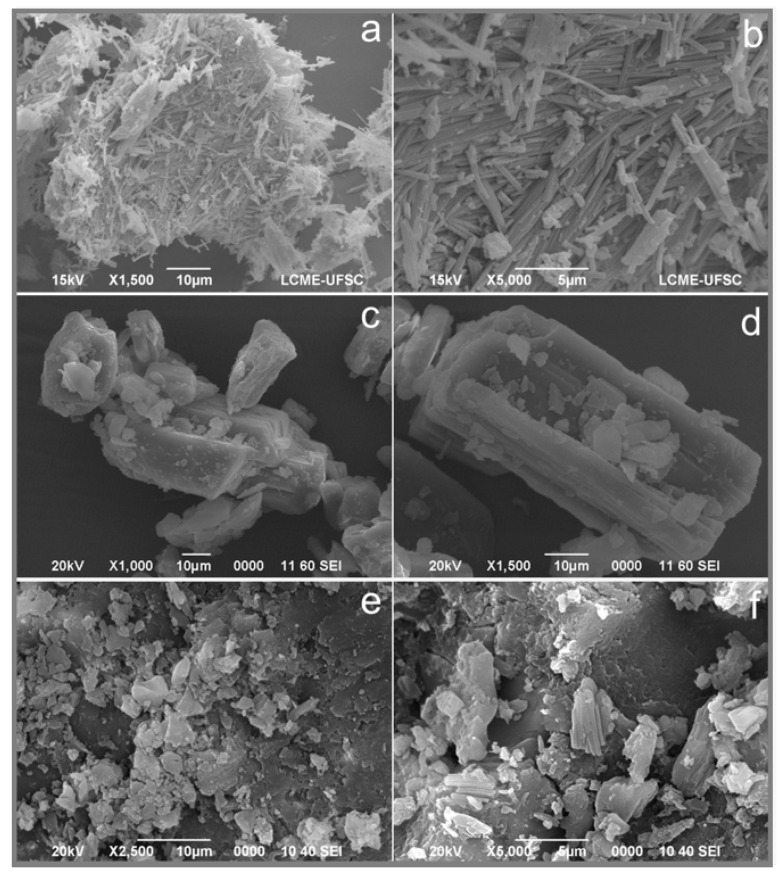
Micrographs of (**a**) SQV 1500× and (**b**) SQV 5000×; (**c**) Piperine (PIP) 1000× and (**d**) PIP 1500×; and eutectic composition (**e**) SQV-PIP 2500× and (**f**) SQV-PIP 5000×.

**Figure 9 pharmaceutics-15-02446-f009:**
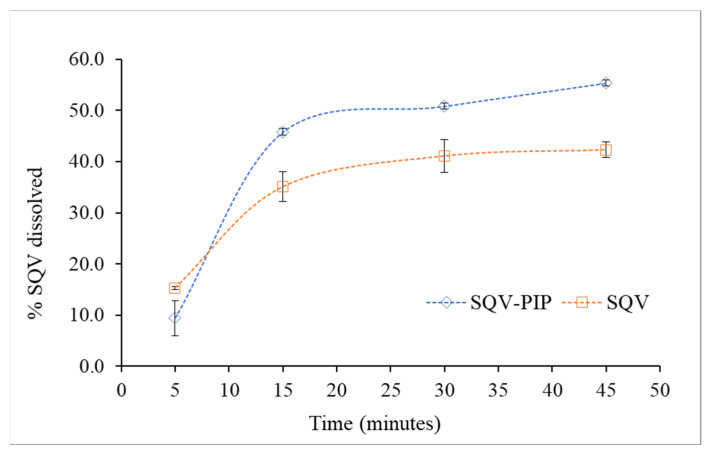
Comparative dissolution profile of the drug SQV and the eutectic mixture SQV-PIP. The dissolution experiment was performed in triplicate (*n* = 3).

## Data Availability

The data presented in this study are available upon request from the corresponding author.

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
