# Peer review of "Saquinavir-Piperine Eutectic Mixture: Preparation, Characterization, and Dissolution Profile"

_pharmaceutics, 2023, doi:10.3390/pharmaceutics15102446_

Round 1
Reviewer 1 Report
This paper reported the preparation, characterization and dissolution profile of Saquinavir-piperine (SQV-PIP) eutectic mixture. SQV was able to form a eutectic system with PIP through liquid assisted grinding approach. This study concludes that the dissolution rate of SQV can be effectively improved through this approach using PIP as conformer. Besides, there are some questions need to be solved.
1. The title should be “The preparation, characterization and dissolution profile of Saquinavir-piperine eutectic mixture”. Besides, in line 29, “FT-IR spectra did not evidence molecular interaction in the solid-state” should be “FT-IR spectra showed no molecular interaction in the solid-state” .
2. Why provided so different magnifications in Figure 8? Please change into the uniformed magnifications.
3. In some figures, the number of experiments replicated need to be added. E.g. n=…
4. In page 6, there are some mistakes in “Figure 3. (a) DSC curves used to construct (b) the eutectic phase diagram where filled and empty triangles represent the variable liquidus line, and filled circles represents the solidus line). (c) Tammann diagram of SQV-PIP system.” There is some inconsistency between the figures and the notes of figures.
5. The letter “SQV” is not shown exactly in the Figure 7.
6. All the references need to be reformatted.
7. Some sentences are described unclearly. For example: " bulk material of the SQV-PIP system in its exact composition 0.6-0.40 was prepared to perform …… Fourier transform infrared spectroscopy (FT-IR), scanning electron microscopy (SEM) and dissolution rate” in line 22-25; " The crystals of SQV-PIP eutectic system show particles with no-well defined shape, as well as crystals in a small particle size due to the grinding process” in line 375-376. Please modify them carefully.
English language should be improved.
Author Response
Reviewer 1
This paper reported the preparation, characterization, and dissolution profile of Saquinavir-piperine (SQV-PIP) eutectic mixture. SQV was able to form a eutectic system with PIP through liquid assisted grinding approach. This study concludes that the dissolution rate of SQV can be effectively improved through this approach using PIP as conformer. Besides, there are some questions need to be solved.
- The title should be “The preparation, characterization and dissolution profile of Saquinavir-piperine eutectic mixture”. Besides, in line 29, “FT-IR spectra did not evidence molecular interaction in the solid-state” should be “FT-IR spectra showed no molecular interaction in the solid-state”.
Thank you for your suggestions. Line 29 (Line 32-33 in the present edited version) has been rewritten as per your suggestion. We have conferred with our co-authors and we would like to ask you to please keep the title. We are agreeing and making all other changes you are suggesting.
- Why provided so different magnifications in Figure 8? Please change into the uniformed magnifications.
Thank you for noticing this. As has been suggested, images with uniformed magnification have been chosen for Figure 8. Magnifications of 1,500 and 5,000X were used for SQV while for SQV-PIP values of 2,500 and 5,000X were selected. SQV and SQV-PIP present smaller crystals in comparison to PIP crystals, which presents considerably larger crystal size than SQV and SQV-PIP crystals, images with magnifications of 1,000 and 1,500X were utilized for PIP. This has been clarified in the manuscript in lines between 400 to 403.
- In some figures, the number of experiments replicated need to be added. E.g. n=…
Thank you for your advice. Numbers of experiments has been included in figures 3, 4 and 9.
- In page 6, there are some mistakes in “Figure 3. (a) DSC curves used to construct (b) the eutectic phase diagram where filled and empty triangles represent the variable liquidus line, and filled circles represents the solidus line). (c) Tammann diagram of SQV-PIP system.” There is some inconsistency between the figures and the notes of figures.
Thank you for noting this. Figure 3 caption has been corrected according to your suggestion.
- The letter “SQV” is not shown exactly in the Figure 7.
The Figure 7 has been changed according to your suggestion.
- All the references need to be reformatted.
All the references were reformatted according to Pharmaceutics journal rules.
- Some sentences are described unclearly. For example: " bulk material of the SQV-PIP system in its exact composition 0.6-0.40 was prepared to perform …… Fourier transform infrared spectroscopy (FT-IR), scanning electron microscopy (SEM) and dissolution rate” in line 22-25; " The crystals of SQV-PIP eutectic system show particles with no-well defined shape, as well as crystals in a small particle size due to the grinding process” in line 375-376. Please modify them carefully.
Thank you for noting this. Sentences have been clarified in the manuscript.
Reviewer 2 Report
The authors have prepared a saquinavir-piperine eutectic mixture to improve the dissolution profile of the drug. This topic will be of interest to readers if it includes adequate experiments and is well explained to help understand the solubility mechanism, toxicological profiles, and intermolecular interactions between the coformer and the drug in the eutectic system. However, there is a lack of data and explanation about the drug-based eutectic mixture in pharmaceutical applications. I believe that this paper would be premature as an article in Pharmaceutics in its current form.
- The main finding of this study is that the saquinavir-based eutectic mixture showed an advantage over the free drug in terms of cumulative dissolution. The authors should discuss the role of the coformer in the eutectic mixture for achieving higher dissolution, as readers may not understand which coformers should be used and how we can manipulate the physicochemical and biopharmaceutical properties of drug molecules using these coformers. I think the authors should clearly present the novelty of their study and its potential applications in the field of pharmaceuticals.
- The authors mentioned that the drug in the eutectic mixture dissolved at a higher rate than the pure drug. The authors need to provide a clear explanation for this enhancement in dissolution.
- The authors used ethanol to prepare the eutectic mixture of the drug. How did the authors confirm the absence of ethanol in the mixture, considering that ethanol is a solubilizing agent?
- Can the authors discuss the intermolecular interactions between the coformer and the free drug, or provide a meaningful explanation for the lower temperature of the eutectic mixture compared to the starting materials in DSC? This would help readers gain a better understanding.
- The authors used a series of coformers as mentioned in lines 18-19 to prepare the eutectic mixture, yet they only explored the saquinavir-piperine eutectic mixture. Why? It is recommended that the authors revise this section for clarity, as a well-stated and informative abstract can help readers quickly grasp the content.
- To confirm the preparation of the eutectic mixture, the authors should provide NMR spectra to demonstrate the stoichiometric ratio between the drug and the coformer. Additionally, it would be helpful to include information about the yields and purities of the eutectic mixtures.
- The authors did not address the toxicological aspects of their study, which is important for evaluating the pharmaceutical application.
Author Response
Reviewer 2
The authors have prepared a saquinavir-piperine eutectic mixture to improve the dissolution profile of the drug. This topic will be of interest to readers if it includes adequate experiments and is well explained to help understand the solubility mechanism, toxicological profiles, and intermolecular interactions between the coformer and the drug in the eutectic system. However, there is a lack of data and explanation about the drug-based eutectic mixture in pharmaceutical applications. I believe that this paper would be premature as an article in Pharmaceutics in its current form.
- The main finding of this study is that the saquinavir-based eutectic mixture showed an advantage over the free drug in terms of cumulative dissolution. The authors should discuss the role of the coformer in the eutectic mixture for achieving higher dissolution, as readers may not understand which coformers should be used and how we can manipulate the physicochemical and biopharmaceutical properties of drug molecules using these coformers. I think the authors should clearly present the novelty of their study and its potential applications in the field of pharmaceuticals.
Thank you for the advice. Further discussion has been included in the manuscript in lines 425-430 and 440-454.
- The authors mentioned that the drug in the eutectic mixture dissolved at a higher rate than the pure drug. The authors need to provide a clear explanation for this enhancement in dissolution.
Thank you for the advice. Further discussion has been included in the manuscript in lines 431-436.
- The authors used ethanol to prepare the eutectic mixture of the drug. How did the authors confirm the absence of ethanol in the mixture, considering that ethanol is a solubilizing agent?
Thank you for noticing this. This has been clarified in the methodology in lines 116-117 and 143-144.
- Can the authors discuss the intermolecular interactions between the coformer and the free drug, or provide a meaningful explanation for the lower temperature of the eutectic mixture compared to the starting materials in DSC? This would help readers gain a better understanding.
Thank you for the advice. Discussion has been included in the manuscript in lines 211-214, 217-222 and 252-255.
- The authors used a series of coformers as mentioned in lines 18-19 to prepare the eutectic mixture, yet they only explored the saquinavir-piperine eutectic mixture. Why? It is recommended that the authors revise this section for clarity, as a well-stated and informative abstract can help readers quickly grasp the content.
Thank you for the advice. This has been clarified in lines 22, and 220-222.
- To confirm the preparation of the eutectic mixture, the authors should provide NMR spectra to demonstrate the ratio between the drug and the coformer. Additionally, it would be helpful to include information about the yields and purities of the eutectic mixtures. stoichiometric
Thanks for your advice. Our Binary Phase and Taman diagrams showed the SQV:PIP eutectic mixture 0.4:0.6 molar ratio (section 3.2, Figure 3b and 3c), allowing us to confirm this data, and we do not have access to solid-state NMR as solution NMR will not give useful information about stoichiometric of this type of materials.
- The authors did not address the toxicological aspects of their study, which is important for evaluating the pharmaceutical application.
Thank you for your important observation. In this regard, an explanation has been added in lines 450-454.
Round 2
Reviewer 2 Report
Thank you for updating the revised manuscript.